# Toric Aberrometric Extended Depth of Focus Intraocular Lens: Visual Outcomes, Rotational Stability, Patients’ Satisfaction, and Spectacle Independence

**DOI:** 10.3390/jpm15030088

**Published:** 2025-02-26

**Authors:** Erika Bonacci, Camilla Pagnacco, Marco Anastasi, Alessandra De Gregorio, Giorgio Marchini, Emilio Pedrotti

**Affiliations:** 1Department of Engineering for Innovation Medicine, Ophthalmology Clinic, University of Verona, 37134 Verona, Italy; erika.bonacci@univr.it; 2Department of Surgery, Dentistry, Paediatrics and Gynaecology, Ophthalmology Clinic, University of Verona, 37134 Verona, Italy; 3Royal Free London NHS Foundation Trust, London NW32QG, UK; marco.anastasi1@nhs.net; 4Ophthalmic Unit, San Bassiano Hospital, 36061 Bassano del Grappa, Italy

**Keywords:** Toric EDOF IOL, rotational stability, spectacle independence

## Abstract

**Objective:** To evaluate visual outcomes, rotational stability, patients’ satisfaction, and spectacle independence after bilateral Toric extended depth of focus intraocular lens (EDOF IOL) implantation. **Methods**: Prospective observational study including cataract patients with bilateral corneal astigmatism between 0.75 and 3.00 D implanted with Toric EDOF IOLs. After three months distance corrected and uncorrected visual acuity at 4 m (DCVA and UDVA), 80 cm (DCI80VA and UI80VA), 67 cm (DCI67VA and UI67VA), and 40 cm (DCNVA and UNVA), IOL stability by Toric IOL Assistant tool (Osiris T, CSO, Florence, Italy), binocular defocus curves, contrast sensitivity (CS), halometry, reading performance, and subjective and objective (Root mean square-RMS, modulation transfer function-MTF, cut-off and point-spread-function-PSF-Strehl ratio) visual quality were evaluated. **Results**: Forty eyes from 20 astigmatic patients were enrolled. Mean refractive spherical equivalent and residual cylinder were −0.21 ± 0.74 D and 0.29 ± 0.31 D, respectively. No patients needed additional surgery due to IOL rotation. Binocular UDVA, UI80VA, UI67VA, and UNVA ≤ 0.2 logMAR was found in 90%, 95%, 85%, and 80%. Distance-corrected visual outcomes have overall shown higher performances. All visual acuities at defocus curves were ≤0.125 logMAR between +0.50 D and −2.00 D. PSF-Strehl ratio, MTF cut-off, RMS were 0.26 ± 0.28, 19.82 ± 12.35, 0.31 ± 0.17. Reading analysis reached 125.42 ± 27.21 words/minute, 92.56 ± 7.82, 0.17 ± 0.15 logMAR and 0.50 ± 0.11 logRAD for mean reading speed, visual acuity score, reading acuity, and critical print size, respectively. CS was higher in photopic conditions. Subjective spectacle independence was achieved in 80% of patients. **Conclusions**: Toric EDOF IOL showed rotational stability and reliable astigmatic correction. It provided spectacle independence and good performance from distance to near distance, reaching high patient satisfaction without undermining binocular quality of vision.

## 1. Introduction

The implementation of presbyopia-correcting premium intraocular lenses (pIOLs) has transformed cataract extraction and intraocular lens implantation into a procedure that not only restores vision but also enhances refractive outcomes, enabling reduced dependence on corrective eyewear [1].

Safety and effectiveness of aberrometric extended depth of focus (EDOF) IOL in terms of visual outcomes, image quality, and spectacle independence, especially in both distance and intermediate visions, have already been reported in both emmetropic and myopic patients, also by this research group [2,3,4,5,6,7].

However, about 40% of the cataract population has a corneal toricity ≥0.75 diopter (D) and 15–20% ≥ 1.25 D [8,9,10,11], leading to insufficient postoperative visual outcomes and patient dissatisfaction [12,13]. Indeed, residual astigmatic refractive errors, arising from preexisting corneal astigmatism or surgical-induced astigmatism, can lead to poor vision, deteriorate image quality and therefore patient dissatisfaction especially after pIOLs implantation [14,15,16,17,18]. For these reasons, the correction of astigmatism is mandatory for the patient’s full satisfaction in pIOL surgery.

However, the use of EDOF IOLs is not without controversy. Some patients report experiencing visual disturbances such as glare and halos, and the IOL effectiveness can be significantly influenced by its rotational stability [19,20].

This topic has raised great interest and some IOLs have already been evaluated [21,22,23,24,25], but studies on these Toric aberrometric EDOF IOL performances are lacking.

The objective of the present study was to assess visual acuity, IOL stability, overall patient satisfaction, and spectacle independence after bilateral implantation of Toric aberrometric EDOF IOL in astigmatic patients.

## 2. Materials and Methods

This prospective, single-center observational clinical study involved astigmatic patients enrolled after cataract surgery with bilateral EDOF Toric IOL (Mini Well Toric, SIFI S.p.A., Catania, Italy) implantation, within 1 week.

This study adhered to the tenets of the Declaration of Helsinki and was approved by the local institutional review board. Written informed consent was obtained from all patients.

Inclusion criteria were bilateral successful cataract surgery performed without complication from the same expert cataract surgeon (EP) in patients with preoperative bilateral corneal regular astigmatism between 0.75 D and 3.00 D, 18 years of age or older with axial length between 22 and 23 mm. Exclusion criteria were presence or history of any ophthalmic disease and/or abnormality such retinal detachment, corneal irregular astigmatism, any previous ocular surgery including corneal or refractive surgery, amblyopia, chronic or recurrent uveitis, active ocular pathology, rubeosis iridis, glaucoma, pseudoexfoliation syndrome, any optic nerve injuries, congenital or acquired color discrimination and contrast sensitivity (CS) reduction and any occurrence of intraoperative complications.

### 2.1. Clinical Protocol

All patients underwent inclusion and exclusion criteria verification checking and enrollment the day after the surgery in the second eye. The IOL stability evaluation was conducted at 1, 7 days after surgery in order to allow possible IOL re-centration and it was re-checked at 1- and 3-months follow-ups. The IOL was considered stable if the angle between the obtained and intended IOL axis resulted <5° and if it did not affect the maximum refraction obtainable up to 0.75 D cylinder, calculated through the Toric IOL Assistant tool (Osiris T, CSO, Florence, Italy) [26].

Complete ophthalmological examination was performed at 3 months follow-up and included binocular and monocular uncorrected distance visual acuity (UDVA) and distance-corrected visual acuity (DCVA) at 4 m, uncorrected and distance corrected at 80 cm intermediate visual acuity (UI80VA and DCI80VA), uncorrected and distance corrected at 67 cm intermediate visual acuity (UI67VA and DCI67VA), uncorrected and distance corrected at 40 cm near visual acuity (UNVA and DCNVA) by means of CSO Vision Charts V14.0 (CSO, Florence, Italy), binocular defocus curve, CS testing (cpg) under photopic (80 cd/m^2^), mesopic (6 cd/m^2^), and scotopic (3 cd/m^2^) light conditions (CSV 1000 HGT; Vector Vision, Greenville, OH), ocular optical quality analysis by Pyramidal WaveFront-based sensor aberrometer (Osiris T Aberrometer, CSO, Florence, Italy) and by a Double-pass aberrometer (Optical Quality Analysis System [OQAS]-Visiometrics SL, Terrassa, Spain), binocular mean reading speed (MRS), visual acuity score (VAS), the critical print size (CPS), and the reading acuity (Mobile App Reading speed test; Aston University, Birmingham, UK), halo test (Aston Halometer, Apple, Cupertino, CA, USA) evaluated by the mean distance from the identified letter to the central LED stimulus, posterior capsular opacity (PCO) evaluation, and the National Eye Institute Refractive Error Quality of Life Instrument 42 (NEI-RQL-42) questionnaire [27]. The questionnaire consists of 13 subscales with 42 items in 16 different question/response category formats. Each item depends on multiple questions within the questionnaire which are converted to a 0 to 100 possible range, following the NEI-RQL-42 user’s manual, so that the lowest and highest possible scores are set at 0 and 100, respectively. All items are scored so that a high score represents better quality of life [28]. Each response, given independently and subjectively by the patient, corresponds to a pre-assigned and validated score.

In case of PCO grade 3 or above (According to Congdon’s study) [29], it was treated by neodymium-doped yttrium aluminum garnet (YAG) laser-capsulotomy, and the 3 months’ evaluation was re-performed after 14 days.

The binocular defocus curve was obtained adding 0.50 D steps from +1.50 to −3.50 D to 4 m DCVA, recording visual acuity for each step. In order to avoid memory effects, presenting letter sequences were randomized and patient’s eyes were occluded between each lens presentation [30].

The variables of the ocular optical quality detected by the Osiris T were total Root Means Square (RMS) and point-spread-function–Strehl ratio (PSF–Strehl ratio) which is defined as the ratio of the peak of the eye’s image intensity from a point source compared to the maximum attainable intensity for an ideal eye limited only by diffraction [31].

The modulation transfer function cut-off (MTF cut-off) was measured by the OQAS, representing the highest spatial frequency (cycles per degree [cpd]) discernible by the ocular optical system.

Binocular reading speed was assessed using a mobile application-based reading speed test on an iPad4 tablet (Apple, Cupertino, CA, USA). The evaluation utilized the Italian version of the Radner Reading Charts. Patients were required to read sentences displayed sequentially in 0.1 logMAR steps, beginning at 1.0 logMAR (Snellen 20/200) and finishing at −0.1 logMAR (Snellen 20/16), or until they selected the “Cannot Read” button. The application continuously recorded the patient’s voice and reading duration to determine the MRS (words/minute), VAS, CPS, and reading acuity (logMAR) [32]. For all charts, CPS was defined as the smallest print size the patients were able to read with the maximum reading speed (logRAD—logarithm of the reading acuity determination). The CPS was obtained by the graph of the reading speed based upon text size in the final printout.

The halo test aimed to quantify, in degrees, the extent to which a glare source obscures a target. The halometer comprised a central light source (LED, Golden Dragon Pluc LCW W5AM.PC, 5000 K color temperature; Osram Licht AG, Munich, Germany) situated in the center of an iPad4 tablet; on-screen letters of 0.3 logMAR (Snellen 20/40) were systematically shifted towards the glare source in 0.05-degree steps using a remote-controlled iPhone 6 application. To determine the halo area, patients were positioned 2 m from the halometer in a dark room and instructed to sequentially identify letters presented in six orientations, spaced 60° apart [33,34].

### 2.2. IOL

The Mini Well Toric IOL is a single-piece, EDOF aspheric preloaded lens based on aberrometric technology. It features four haptics and a double-square-edge, with a total diameter of 10.75 mm and an optical zone diameter of 6 mm. The lens is structured into three optical zones: the inner (approximately 1.8 mm wide) and middle zone (approximately 3.0 mm wide) exhibit spherical aberrations of opposite signs. The central zone (positive spherical aberration) is designed to add power for the near vision and to contribute to broaden the defocus in a multiplicity of foci widening the peak from near to intermediate and giving the lens its peculiar multifocality. The intermediate zone (negative spherical aberration) increases the focus depth in the far field region and intermediate vision. The outer zone is a monofocal zone that provides the far focus. The three optical zones are seamlessly connected by an “active transition zone” without gaps between them.

The Mini Well Toric IOL has a spherical equivalent dioptric range from +7.00 to +30.00 D in 0.50 D increments and Toric power from 1.00 D to 4.50 D with 0.50 D increments.

The IOL, designed to be inserted into the capsular bag through corneal micro-incisions, is made of an ultraviolet blue-light filter copolymer.

### 2.3. Statistical Analysis

Statistical analysis was performed using IBM SPSS software version 24 for MacIntosh (IBM-SPSS). The Shapiro–Wilk test was used to determine data distribution. The sample size was calculated based on considering that 2 D of astigmatism induces near visions’ impairment of 0.5 logMar and hypothesizing a DCNVA of the EDOF IOL of 0.01 ± 0.10 [5]. We detected that 18 patients would ensure a power of 90% and a significance (alpha) of 1%. We enrolled 20 patients to avoid biases in sample size due to lost follow-ups. Postoperative data are presented at 3 months. Categorical variables are presented as frequencies and percentages. All quantitative results are reported as mean ± standard deviation for parametric distribution and as median ± interquartile range for non-parametric distribution.

## 3. Results

Forty eyes from 20 astigmatic patients were enrolled. The mean age of the subjects enrolled was 73.5 ± 4.24 years (median: 74 years, range: 65 to 81 years). The average spherical dioptric power of the implanted IOL was 22.32 ± 1.77 D (median: 22.75 D, range: 20.00 to 25.50 D) with a mean Toric power of 2.94 ± 1.30 D (median: 2.50 D, range: 1.00 to 4.50 D) (Table 1). None of the patients showed IOL decentration or rotation requiring additional surgery throughout the entire study period: at the 3 months’ follow-up visit, the mean rotation was 2.5 ± 2° (range min–max 0–4.3°) and 20 implanted IOLs (50%) showed no rotation.

At the 3-month follow-up examination, PCO at stage 3 and 4 was found in four eyes (10%), which underwent YAG laser-capsulotomy and re-examination.

### 3.1. Visual Outcomes

The mean postoperative refractive spherical equivalent was −0.21 ± 0.74 D (median: −0.13 D; range: −1.50 to +1.00 D). It was within ± 1.00 D in 95% and between ±0.50 D in 67.5%. The mean postoperative refractive cylinder was 0.29 ± 0.31 D (median: 0.25 D, range: 0.00 to 1.00 D). It was under 0.50 D in 87.5% of cases and within 0.25 D in 62.5% of cases (Figure 1).

Table 2 shows the postoperative monocular and binocular visual and refractive data.

Figure 2 illustrates the cumulative monocular and binocular visual acuity outcomes.

Monocular UDVA, UI80VA, UI67VA, and UNVA of 0.2 logMAR (Snellen 20/32) or better was found in 75%, 87.5%, 75%, and 72.5% of eyes, respectively. Correspondingly, binocular UDVA, UI80VA, UI67VA, and UNVA of 0.2 logMAR (Snellen 20/32) or better was achieved in 90%, 95%, 85%, and 80% of cases, respectively. Distance-corrected visual outcomes were slightly better in all categories, with the exception of monocular DCI80VA that performed similarly to UI80VA. Differences between uncorrected visual acuities and DCVA (in both monocular and binocular conditions) for all evaluated distances are shown in Figure 3. Specifically, the figure represents the percentages of eyes having the same logMAR value of uncorrected visual acuity compared to the logMAR value obtained with DCVA, in blue when evaluated monocularly and in orange when evaluated binocularly. Also, it shows the percentages of eyes having better or worse uncorrected visual acuities and the entity of the difference compared to their DCVA counterparts. Monocular UNVA had the same logMAR value as DCNVA in 62.5% of the eyes, and UNVA was within 0.1 logMAR of DCNVA in 82.5% of the eyes. Binocular UNVA was better or equal to DCNVA in 60% of the patients and it was within 0.1 logMAR of DCNVA in 85%.

### 3.2. Defocus Curve Outcomes

Figure 4 illustrates the mean binocular defocus curve. All visual acuities were found lower or equal to 0.125 logMAR between +0.50 and −2.00 D showing a very slow decline through the negative defocus. No statistical differences were found between −1.50 D and −2.00 D (*p* = 0.32) nor between −1.50 D and −2.50 D (*p* = 0.06).

### 3.3. Contrast Sensitivity (CS) Outcomes

Figure 4 presents the binocular CS function measured under scotopic, mesopic, and photopic light conditions. Also, it reports the CS values of the same IOL platform in its monofocal version. The CS under photopic condition was statistically better than the CS measured in both scotopic and mesopic conditions (*p* < 0.05). Moreover, the values of the CS under mesopic conditions were better compared to the ones obtained under scotopic conditions (*p* < 0.05).

### 3.4. Ocular Optical Quality Outcomes

The mean postoperative MTF cut-off detected by OQAS was 19.82 ± 12.35 cpd (median: 17.44 cpd; range: 0.05–53.15 cpd).

The mean postoperative PSF–Strehl ratio and ocular RMS collected through Osiris T were 0.26 ± 0.28 (median: 0.13; range: 0.08–0.58), 0.31 ± 0.17 (median: 0.24; range: 0.11–0.60), respectively.

### 3.5. Halometry

Figure 4 presents the mean cut-off values. The mean distance from the halo LED source was 0.71 ± 0.36 (median 0.6; range: 0.20–2.00) for all six vertices.

### 3.6. Reading Analysis

The MRS was 125.42 ± 27.21 words/minute (median: 128.1 words/minute; range: 67.97–159.35 words/minute). The mean reading acuity was 0.17 ± 0.15 logMAR (median: 0.1 logMAR; range: 0.00–0.40 logMAR). The mean VAS was 92.56 ± 7.82 (median: 91.6; range: 80.00–100.00). The mean CPS was 0.50 (± 0.11) logRAD.

### 3.7. Quality of Life Outcomes

Table 3 summarizes the postoperative quality of life outcomes for the 13 parameters assessed using the NEI RQL-42 questionnaire. The best results were obtained in the activity limitation and far vision categories. Subjective spectacle independence was achieved in 80% of them (items 13, 14, 15, and 16). Moreover, glares were not perceived in 45% of the patients and 55% of them reported it as occasional (items 17 and 38 b).

## 4. Discussion

In this study, the Miniwell Toric EDOF IOL showed successful visual restoration, good quality of vision, and high stability after implantation. All monocular and binocular visual acuities achieved very high performance at all distances also showing elevated levels of predictability and spectacle independence. This is particularly relevant as patients undergoing pIOL surgery aim for spectacle independence and the predictability is central in this surgery.

The spectacle independence is not only subjectively declared by patients through the questionnaire (the majority of the subjects declared not to need glasses for far nor near vision) at NEI RQL-42 and also noted objectively when evaluating visual acuity at various distances and at the defocus curve, but the majority of patients have no visual discrepancies between the DCVA and the uncorrected visual acuity, especially in binocular vision (Figure 3) also indicating high reproducibility in the use of the IOL. Few patients had slightly worse vision than DCVA and these were those whose astigmatic residual was above 0.75. In these cases, the cylinder of the posterior surface of the cornea has affected the calculation of the IOL.

The obtained spectacle independence is supported also by the high performance in terms of MRS and mean reading acuity.

These results are overall comparable to those of the same non-Toric IOL with which it shares good performances with regard to visual acuity and quality of vision. Certainly, IOL high rotational stability allows not only to correct corneal astigmatism, preventing this from affecting the performance of the IOL, but also it prevents the addition of new aberrations in the ocular wavefront, which are inevitable following rotation of the IOL.

In fact, none of our patients needed a second surgery to correct IOL rotation. This is probably due to the shape of the IOL: the plate-haptic Toric IOLs permit a lower rotation rate compared to loop-haptic ones [35,36]. Even if other studies did not collect postoperative aberrometric measurements which limits comparison between data, the mean rotation was lower in this IOL compared to the three months’ results from Georgiev et al. which were 3.5° ± 3.5° [24] and the overall rotation was lower than 5 degrees, similarly to the data reported from other authors [21,22,23]. Also, in our study, the postoperative refractive astigmatism was lower than 0.75 D in 92.5% which is better than what was found for the enVista Toric IOL where this outcome was achieved in 88.1% of patients [24]. Various factors contribute to the rotational stability of Toric IOLs, including biological parameters, surgical techniques employed during implantation, and the design of the IOLs which can be categorized into one-piece or three-piece designs, with haptic configurations available in C-loop or plate designs [37,38]. The shape of these lenses is critical for ensuring proper alignment and rotational stability post-implantation [22]. New designs are continually emerging to improve the effectiveness of Toric IOLs. The Mini Toric Ready lens, introduced by SIFI S.p.A., features fenestrated haptics that enhance contact with the capsular bag’s equator, facilitating better rotational stability compared to traditional designs [39,40].

Although a direct comparison with the same EDOF non-Toric platform already studied by this research group [2,5] is lacking, visual evaluations showed comparable results for distance visual acuities. However, an unexpected visual difference was notable in binocular UNVA and DCNVA, which showed higher performance in the Toric implantation. This behavior was also found at the defocus curve. The latter, indeed, showed higher performances in the Toric platform between −2.00 and −2.50 D. Otherwise, the EDOF non-Toric IOL performed a little better for defocus values equal to −1.50 and for hyperopic values. Furthermore, comparing the Miniwell Toric defocus curve with the Toric Tecnis Symfony IOL targeted for emmetropia reported by Sandoval et al. [38], the Miniwell Toric IOL shows better performances between −2.00 and −3.00 D.

As for contrast sensitivity, the values achieved in photopic conditions without glare share the same behavior as the AcrySof^®^ IQ studied by Bala et al. [41], while the Miniwell Toric showed better PSF–Strehl ratio values compared to the ones achieved by the Symfony IOL [42].

Similarly to what was found on a previous multicentric study [43], overall patient satisfaction was achieved in 80% of the patients. The remaining 20% were just somewhat dissatisfied most likely due to reported symptoms such as tearing and itching (NEI RQL-42 questions 18 and 36). Furthermore, the same patients that were not fully satisfied declared not to experience any activity limitation (NEI RQL-42 questions 12, 33, 34, and 35) showing that this outcome is most likely due to undiagnosed dry eye disease which often becomes manifest after cataract surgery [44].

In conclusion, Mini Well Toric EDOF IOLs proved to be stable and allow high functional outcomes, spectacle independence, and satisfying quality of vision.

## Figures and Tables

**Figure 1 jpm-15-00088-f001:**
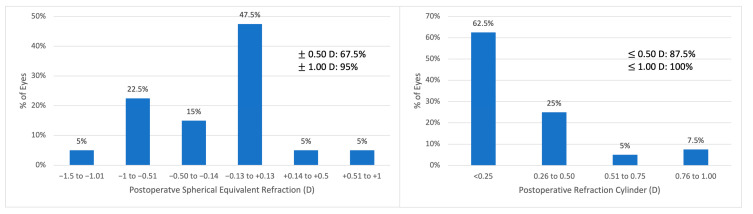
Postoperative spherical equivalent refractive accuracy and postoperative refractive cylinder. EDOF = extended depth of focus; D = diopter.

**Figure 2 jpm-15-00088-f002:**
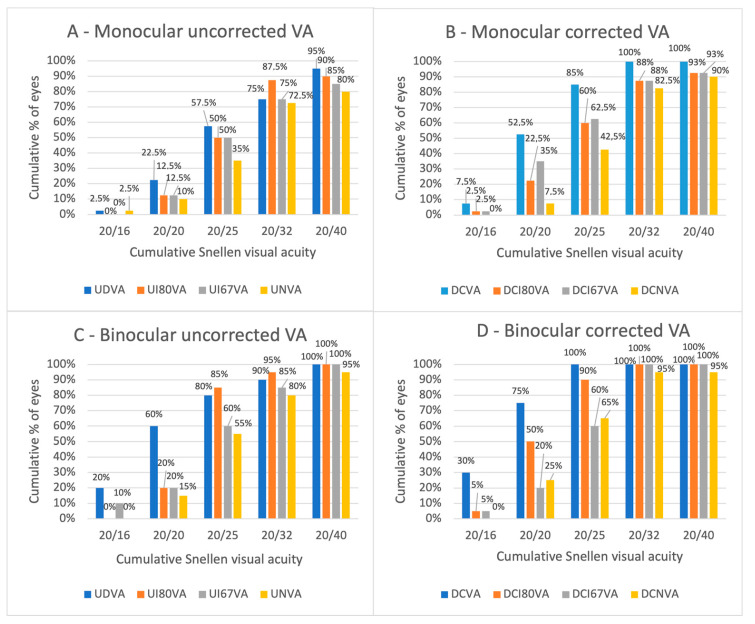
Cumulative distribution of postoperative monocular (**A**,**B**) and binocular (**C**,**D**) visual outcomes for the Toric extended depth of focus (EDOF) Miniwell Toric intraocular lens (IOL). UDVA = uncorrected distance visual acuity; UI80VA = uncorrected intermediate (80 cm) visual acuity; UI67VA = uncorrected intermediate (67 cm) visual acuity; UNVA = uncorrected near visual acuity; DCVA = corrected distance visual acuity; DCI80VA = distance-corrected intermediate (80 cm) visual acuity; DCI67VA = distance-corrected intermediate (67 cm) visual acuity; DCNVA = distance-corrected near visual acuity.

**Figure 3 jpm-15-00088-f003:**
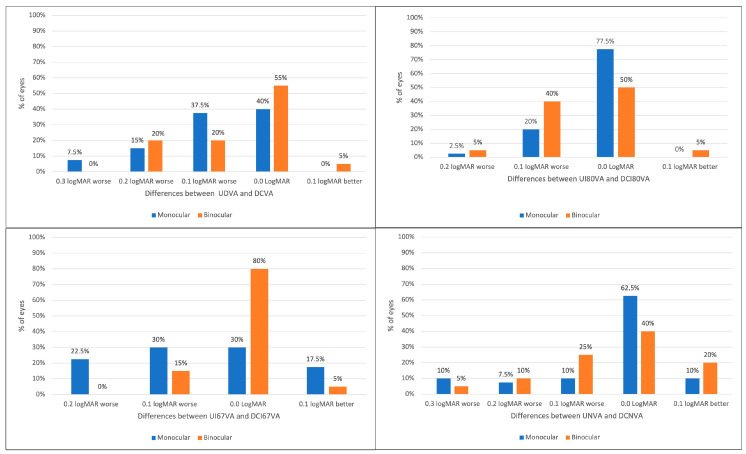
The graph shows the differences in logMAR between uncorrected visual acuity and DCVA, collected monocularly (blue) and binocularly (orange) at different distances. Monocular and binocular uncorrected (UDVA) versus corrected (DCVA) distance visual acuity. Monocular and binocular uncorrected (UI80VA) versus distance-corrected (DCI80VA) intermediate at 80 cm visual acuity. Monocular and binocular uncorrected (UI67VA) versus distance-corrected (DCI67VA) intermediate at 67 cm visual acuity. Monocular and binocular uncorrected (UNVA) versus distance-corrected (DCNVA) near visual acuity.

**Figure 4 jpm-15-00088-f004:**
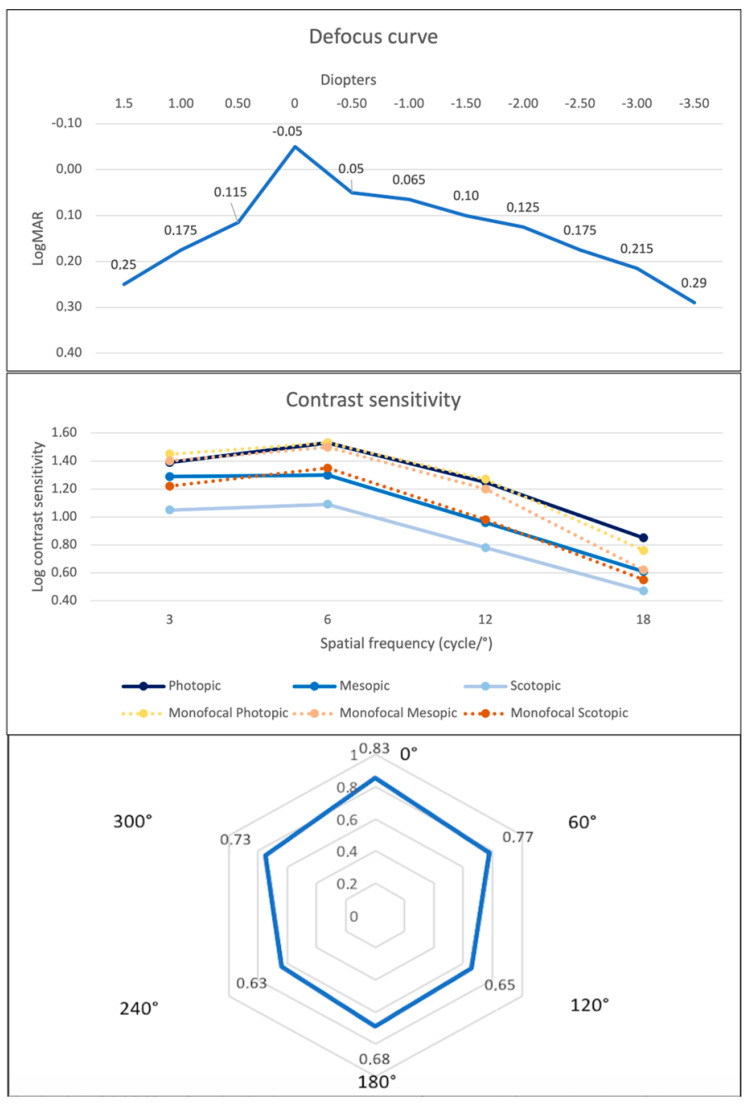
Mean binocular defocus curve, contrast sensitivity function of the EDOF Toric IOL and its monofocal version measured under scotopic, mesopic, and photopic conditions and six-vertex halometry 3 months after bilateral Toric EDOF IOL implantation. D = diopters.

**Table 1 jpm-15-00088-t001:** Demographic data, preoperative corneal astigmatism (D), spheric and Toric IOL powers (D), postoperative refractive spherical equivalent (D), and refractive cylinder (D).

Demographic Data	Mean ± Standard Deviation	Median	Range
Male: Female	6:14	-	-
Age	73.5 ± 4.24 years	74 years	65 to 81 years
Preoperative corneal astigmatism	1.94 ± 0.92 D	1.58 D	0.75 to 3.00 D
IOL spherical power	22.32 ± 1.77 D	22.75 D	20.00 to 25.50 D
IOL Toric power	2.94 ± 1.30 D	2.50 D	1.00 to 4.50 D
Refractive spherical equivalent	−0.21 ± 0.74 D	−0.13 D	−1.50 to +1.00 D
Refractive cylinder	0.29 ± 0.31 D	0.25 D	0.00 to 1.00 D

**Table 2 jpm-15-00088-t002:** Postoperative monocular and binocular visual data. UDVA = uncorrected distance visual acuity; SD = standard deviation; DCVA = corrected distance visual acuity; UI80VA = uncorrected intermediate (80 cm) visual acuity; DCI80VA = distance-corrected intermediate (80 cm) visual acuity; UI67VA = uncorrected intermediate (67 cm) visual acuity; DCI67VA = distance-corrected intermediate (67 cm) visual acuity; UNVA = uncorrected near visual acuity; DCNVA = distance-corrected near visual acuity.

Visual Acuity	Monocular Vision		Binocular Vision	
(logMAR)	Mean ± SD	Median (Range)	Mean ± SD	Median (Range)
UDVA	0.15 ± 0.13	0.10 (−0.10 to 0.50)	0.05 ± 0.12	0.00 (−0.10 to 0.30)
DCVA	0.06 ± 0.08	0.00 (−0.10 to 0.20)	−0.01 ± 0.08	0.00 (−0.10 to 0.10)
UI80VA	0.16 ± 0.13	0.10 (0.00 to 0.60)	0.1 ± 0.07	0.10 (0.00 to 0.30)
DCI80VA	0.14 ± 0.14	0.10 (−0.10 to 0.60)	0.06 ± 0.08	0.05 (−0.10 to 0.20)
UI67VA	0.18 ± 0.15	0.10 (0.00 to 0.60)	0.13 ± 0.12	0.10 (−0.10 to 0.30)
DCI67VA	0.13 ± 0.15	0.10 (−0.10 to 0.60)	0.12 ± 0.09	0.10 (−0.10 to 0.20)
UNVA	0.23 ± 0.18	0.20 (−0.10 to 0.80)	0.16 ± 0.12	0.10 (0.00 to 0.50)
DCNVA	0.19 ± 0.13	0.20 (0.00 to 0.60)	0.13 ± 0.12	0.10 (0.00 to 0.50)

**Table 3 jpm-15-00088-t003:** Postoperative Quality of Life Scores—NEI RQL-42 scores subdivided in 13 categories.

Parameter	Mean ± SD	Median (Range)
Clarity of vision	69.97 ± 19.31	69.79 (31.33 to 100.00)
Expectations	72.50 ± 30.24	75.00 (0.00 to 100.00)
Near vision	51.98 ± 21.29	47.92 (31.25 to 81.25)
Far vision	80.58 ± 18.44	77.50 (56.66 to 100.00)
Diurnal fluctuations	73.75 ± 22.75	79.17 (41.67 to 100.00)
Activity limitations	92.19 ± 12.15	100.00 (68.75 to 100.00)
Glare	76.88 ± 24.76	72.50 (47.50 to 100.00)
Symptoms	58.75 ± 13.88	58.93 (39.29 to 96.43)
Dependence on correction	70.21 ± 19.97	62.50 (41.67 to 100.00)
Worry	49.38 ± 23.11	37.50 (25.00 to 100.00)
Suboptimal correction	89.38 ± 15.32	100.00 (62.50 to 100.00)
Appearance	73.00 ± 24.90	73.33 (20.00 to 100.00)
Satisfaction with correction	67.00 ± 28.49	70.00 (20.00 to 100.00)

## Data Availability

The original contributions presented in this study are included in the article. Further inquiries can be directed to the corresponding author.

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
