# Peer review of "Toric Aberrometric Extended Depth of Focus Intraocular Lens: Visual Outcomes, Rotational Stability, Patients’ Satisfaction, and Spectacle Independence"

_jpm, 2025, doi:10.3390/jpm15030088_

Round 1
Reviewer 1 Report
Comments and Suggestions for Authors
In this paper, the authors evaluate the visual outcomes after bilateral implantation of the Toric EDOF intraocular lens (mini well, SIFI). The authors evaluated a series of parameters in 20 subjects including: corrected and uncorrected visual acuity at far, 80cm, 67cm, 40 cm distances, binocular defocus curves, contrast sensitivity in photopic, mesopic and scotopic range, halometry, reading performance and objective (RMS, MTF and PSF) and subjective (by means of questionnaire) visual quality.
Overall evaluation: The paper presents a huge amount of data related to the implantation of toric EDOF IOL. Nonetheless, to completely understand what have been evaluated and analysed, the introduction and method sections should be improved along with the presentation and analysis of the results. Below are some major comments to improve the manuscript and some minors that the authors should address.
Major comments:
1. The introduction is very poor, in just a few lines. Author must include some more background, and present what the current paper is introducing new to the field. Can the authors give some more details about previous literature on toric IOL performance evaluation?
2. section on materials and methods: authors should include the demographic data of the patients enrolled in the study and their refraction. Part of the information is reported in the results section lines 160-162. There, the authors should add the range of the toric power (only the mean toric power is reported).
3. line 62. the inclusion criteria allow bilateral corneal regular astigmatism between 0.75D to 4.5D. Nonetheless, the mini-well toric IOL has a toric power between 1D and 4.5D. What happens with patients with 0.75D? were they implanted with 1D toric power? It should have been useful to add the individual information of the IOL implanted in each patient.
4. In general, in the clinical protocol section, the authors should describe better what have been evaluated. For example, what the halo test visual acuity score consisted in? How do they evaluate the PCO? How did they analyse the responses to the questionnaire to end up with the table B.What were the fundamental questions used for table B analisis.
5. lines 186-187. Authors mentioned: “Distance corrected visual outcomes were slightly better in all categories, with the exception of monocular DCI80VA (Figure 3)” Authors should explain better this sentence because according to figure 2, there is no such exception.
6. Figure 3 explanation is missing. We do not know what the authors are showing and additionally the figure is cut so part of the data is missing.
7. Evaluation of the defocus outcome, lines 212-213. Authors should explain why they look particularly at those data (No statistical differences were found nor between -1.50 D and -2.00 D (p = 0.32) nor between -1.50 D and -2.50 D (p = 0.06)).
8. Contrast sensitivity outcomes, lines 215-218 and figure 4: the description of the “normal mean CS” is missing. In the figure, it appears that the legend for scotopic and photopic conditions has been inverted
9. Ocular optical quality. Lines 220-224. The authors should explain where those data are coming from.
10. quality of life outcomes, lines 238-246 and table B. The results from the questionnaire should be better explained. One line can be incorporated to explain each parameter from the questionnaire (and in the method section, how was calculated the score between 0 and 100)
Minor comments:
Abstract lines 27-28. Units are missing
Lines 93-94. When explaining the score attributed to the questionnaire, authors said that “All items are scored so that a high score represents better quality of life” but it does not seem true for the “worry” and “symptoms” parameters of table B. Can the authors give more information?
Line 167. “which YAG laser-capsulotomy and re-examination” should be “with” instead of “which”?
Figure 1: the number in the left part does not add 100% (but 100.5%) and the range “+1.01 to +3D” is missing, according to the text lines 169-170. Or is there an error in the text and the mean postoperative refractive spherical equivalent raged from -1.5D to +1D?
Line 171-173 and figure 1 right: can the authors comment on the possible reason why there is 7.5% of the patients with an astigmatic error above 0.75D?
Table A: It would have been interesting to also have the pre op data
Lines 234-237. Please state what are the abbreviations MRS, VAS and CPS
Author Response
Comment 1: The introduction is very poor, in just a few lines. Author must include some more background, and present what the current paper is introducing new to the field. Can the authors give some more details about previous literature on toric IOL performance evaluation?
Reply 1: Thank you for your comment, we were happy to extended our introduction including some recent works on toric IOLs performance evaluation. Furthermore, we added a paragraph on the discussions reporting recent literature on toric IOLs.
- C. Giers et al., “Functional results and photic phenomena with new extended-depth-of-focus intraocular Lens,” BMC Ophthalmol., vol. 19, no. 1, p. 197, Aug. 2019, doi: 10.1186/s12886-019-1201-3.
- [20] Ferrando Gil, A. Churruca Irazola, I. Reparaz, G. Lauzirika, I. Martínez-Soroa, and J. Mendicute, “Visual, Refractive, Functional, and Patient Satisfaction Outcomes After Implantation of a New Extended Depth-of-Focus Intraocular Lens,” Clin. Ophthalmol. Auckl. NZ, vol. 18, pp. 3801–3813, 2024, doi: 10.2147/OPTH.S499911.
- Lin, D. Ma, and J. Yang, “Insights into the rotational stability of toric intraocular lens implantation: diagnostic approaches, influencing factors and intervention strategies,” Front. Med., vol. 11, p. 1349496, 2024, doi: 10.3389/fmed.2024.1349496.
- Pastor-Pascual, P. Orts-Vila, P. Tañá-Sanz, S. Tañá-Sanz, R. Ruiz-Mesa, and P. Tañá-Rivero, “Non-diffractive, toric, extended depth-of-focus intraocular lenses in eyes with low corneal astigmatism,” Eye Vis. Lond. Engl., vol. 11, no. 1, p. 14, Apr. 2024, doi: 10.1186/s40662-024-00380-7.
- P. Sandoval, S. Lane, S. G. Slade, E. D. Donnenfeld, R. Potvin, and K. D. Solomon, “Defocus Curve and Patient Satisfaction with a New Extended Depth of Focus Toric Intraocular Lens Targeted for Binocular Emmetropia or Slight Myopia in the Non-Dominant Eye,” Clin. Ophthalmol., vol. 14, pp. 1791–1798, Jun. 2020, doi: 10.2147/OPTH.S247333.
- Savini, G. Alessio, G. Perone, S. Rossi, and D. Schiano-Lomoriello, “Rotational stability and refractive outcomes of a single-piece aspheric toric intraocular lens with 4 fenestrated haptics,” J. Cataract Refract. Surg., vol. 45, no. 9, pp. 1275–1279, Sep. 2019, doi: 10.1016/j.jcrs.2019.05.015.
Comment 2: section on materials and methods: authors should include the demographic data of the patients enrolled in the study and their refraction. Part of the information is reported in the results section lines 160-162. There, the authors should add the range of the toric power (only the mean toric power is reported).
Reply 2: Thank you for your comment, we have reported a table with the demographic data of our patients, their preoperative corneal astigmatism, IOL spherical and toric power and postoperative refraction at 3 months.
Comment 3: line 62. the inclusion criteria allow bilateral corneal regular astigmatism between 0.75D to 4.5D. Nonetheless, the mini-well toric IOL has a toric power between 1D and 4.5D. What happens with patients with 0.75D? were they implanted with 1D toric power? It should have been useful to add the individual information of the IOL implanted in each patient.
Reply 3: The toric power of the IOL ranges from +1.00 to +4.50 D (IOL plane). This corrects corneal astigmatism from 0.65 D to 3.00 D (corneal plane). Due to optical physics, the toric power of the IOL must necessarily be greater than the corneal astigmatism power. More importantly, I would like to highlight that the typo is in the 4.50 value. The enrolled patients had corneal astigmatism between 0.75 D and 3.00 D.
Comment 4: In general, in the clinical protocol section, the authors should describe better what have been evaluated. For example, what the halo test visual acuity score consisted in? How do they evaluate the PCO? How did they analyse the responses to the questionnaire to end up with the table B. What were the fundamental questions used for table B analisis.
Reply 4: Thank you for this comment which allow us to clarify and rectify.
- The test conducted with the Halometer gives scores which represents the Evaluation of the Disk Halo Size studied by the halometer (Aston Halometer). The visual acuity score is a different kind of measurement performed with the same instrument used for the mean reading speed.
- PCO: After pharmacological dilation of the pupil, the presence of PCO was graded as follows: absent, no opacity or opacity limited to the peripheral capsule; 1+, any wrinkling or opacity of the capsule affecting a circle 4 mm in diameter and centered on the visual axis, but not compromising the view of the posterior pole; 2+, central/paracentral opacity as described above sufficient to degrade details of the macula slightly, but still allowing the cup/disc ratio to be readily ascertained; 3+, central/paracentral opacity as defined above, but sufficient to make ascertainment of the cup/disc ratio difficult; 4+, central/paracentral opacity as defined above, but sufficient to make visualization of fundus details difficult or impossible.
- Table B (now C) NEI RQL 42 QUESTIONNAIRE: The questionnaire used is the NEI-RQL 42, a validated and widely used tool for assessing subjective satisfaction in pseudophakic refractive surgery. I am attaching the user manual, which includes the questions submitted for each evaluated item.
Comment 5: lines 186-187. Authors mentioned: “Distance corrected visual outcomes were slightly better in all categories, with the exception of monocular DCI80VA (Figure 3)” Authors should explain better this sentence because according to figure 2, there is no such exception.
Reply 5: Thank you very much for the note, I agree with the reviewer. The reference was mistakenly placed. I have reworded the sentence, and I hope it is now clearer. (lines 215-216)
Comment 6: Figure 3 explanation is missing. We do not know what the authors are showing and additionally the figure is cut so part of the data is missing.
Reply 6: The figure 3 illustrates the differences in terms of visual acuity (logMAR) between CDVA and UDVA in both monocular and binocular vision at the all evaluated distances in this study. It is particularly relevant as patients undergoing this type of surgery aim for spectacle independence. At the same time, small discrepancies between the expected refractive target and the obtained refractive target are possible. Therefore, the graphs explain both the best IOL performances based on the lens design, by CDVA, and the real-life outcomes achieved under the best possible biometric calculation conditions, by UDVA. The image wasn’t cut, the Y axis was adjusted to better show the graphs but as suggested by the reviewer the Y axis now goes from 0 to 100% for all the 4 graphs making them more comparable. Also, we added some explanation both in the text and in the figure legend to make it more easily comprehensible.
Comment 7: Evaluation of the defocus outcome, lines 212-213. Authors should explain why they look particularly at those data (No statistical differences were found nor between -1.50 D and -2.00 D (p = 0.32) nor between -1.50 D and -2.50 D (p = 0.06)).
Reply 7: Thank you for your question. We focused on those specific diopters in the defocus curve because they assess the lens near vision performance (-1.50 D evaluates 66 cm, -2.00 D evaluates 50 cm, and -2.50 D evaluates 40 cm).
Comment 8: Contrast sensitivity outcomes, lines 215-218 and figure 4: the description of the “normal mean CS” is missing. In the figure, it appears that the legend for scotopic and photopic conditions has been inverted
Reply 8: Thank you for noticing, we rectify the legend in the figure.
In fact, there is no universally defined gold standard for “normal” contrast sensitivity (CS). However, we can illustrate the CS performance for the same IOL platform in its monofocal version. It was not initially included because it was not the focus of this study.
Comment 9: Ocular optical quality. Lines 220-224. The authors should explain where those data are coming from.
Reply 9: Thank you for your comment, we believed it was more suitable to specify it in the clinical protocol but as you suggested we have modified the paragraph to make it more understandable.
Comment 10: quality of life outcomes, lines 238-246 and table B. The results from the questionnaire should be better explained. One line can be incorporated to explain each parameter from the questionnaire (and in the method section, how was calculated the score between 0 and 100)
Reply 10: Thank you for the note. Each item depends on multiple questions within the questionnaire. Each response, given independently and subjectively by the patient, corresponds to a pre-assigned and validated score, which is available in the User Manual but not included in the administered questionnaire. We have specified this in the Methods section. (Lines 101-108)
Minor comments:
- Abstract lines 27-28. Units are missing
Thank you very much for the note. The units were omitted due to word/character count limitations. They have now been added.
- Lines 93-94. When explaining the score attributed to the questionnaire, authors said that “All items are scored so that a high score represents better quality of life” but it does not seem true for the “worry” and “symptoms” parameters of table B. Can the authors give more information?
Sure, they represent a pre-assigned score, which is available in the User Manual.
- Line 167. “which YAG laser-capsulotomy and re-examination” should be “with” nstead of “which”?
Sorry for the mistake. The word "underwent" was missing in the sentence.
- Figure 1: the number in the left part does not add 100% (but 100.5%) and the range “+1.01 to +3D” is missing, according to the text lines 169-170. Or is there an error in the text and the mean postoperative refractive spherical equivalent raged from -1.5D to +1D?
Thank you for noticing, the graph should have reported 22,5% instead of 23% (it was an automatic approximation). We have amended it. As for the range of the refractive spherical equivalent, it is -1.50 to +1.00 D (the +3.00D was a typing mistake), this is why the range +1.01 to +3D was missing from the graph.
- Line 171-173 and figure 1 right: can the authors comment on the possible reason why there is 7.5% of the patients with an astigmatic error above 0.75D?
Thank you for your question. The possible reason to explain why in only two eyes we had an astigmatic error of 1D could be related to the fact that the cylinder of the posterior surface of the cornea may have affected the calculation of the IOL. We added it in our discussions.
Table A: It would have been interesting to also have the pre op data
Thank you for this suggestion, we have created a new table (Table A) which reports further information, unfortunately since patients were recruited after the second surgery, data on preoperative visual acuity were not recorded.
- Lines 234-237. Please state what are the abbreviations MRS, VAS and CPS
The meaning of the acronyms is available on lines 96-97
Reviewer 2 Report
Comments and Suggestions for Authors
The authors present a study about intraocular lens stability after implantation of toric EDOF IOLs. The research seems to be well conducted and the manuscript is well structured. However, I would suggest some changes:
1) 40 eyes of 20 patients were included in the study. This should appear at the very beginning of the results section, not methods.
2) The discussion section should be rewritten enterely. First, a short summary of the main outcomes should be included. Then, the authors should explain the reason that is responsible for those outcomes, and finally they should compare their outcomes to other previous published articles. In case there are none, they should compare with similar IOLs. They need to make these comparisons not only with visual acuity, but with all the other performed tests, as well. All these information is missing in its present form.
Author Response
Comment 1: 40 eyes of 20 patients were included in the study. This should appear at the very beginning of the results section, not methods.
Reply 1: Thank you very much for your comments. You are right; it sounds like a result. Actually, it is a data point we obtained through the statistical evaluation of the sample size. (we have deleted “40 eyes of 20 patients” line 53)
Comment 2: The discussion section should be rewritten enterely. First, a short summary of the main outcomes should be included. Then, the authors should explain the reason that is responsible for those outcomes, and finally they should compare their outcomes to other previous published articles. In case there are none, they should compare with similar IOLs. They need to make these comparisons not only with visual acuity, but with all the other performed tests, as well. All these information is missing in its present form.
Reply 2: Thank you for your suggestion, we have reorganized the discussion and started with a summary of our main outcomes, followed by a possible explanation of the results. Also, as suggested, we implemented our discussion comparing our outcomes to all articles we could find that studied toric EDOF IOLs. Specifically, we were able to find data, not only on rotational stability but also on residual post operative refractive astigmatism, on defocus curve, PSF Strehl ratio and patient satisfaction which we included in the discussion.
Round 2
Reviewer 1 Report
Comments and Suggestions for Authors
The authors responded to each of the comments raised in the first revision and have now provided a revised version that improves the manuscript.
Author Response
The authors responded to each of the comments raised in the first revision and have now provided a revised version that improves the manuscript.
-Thanks for the comments.
Reviewer 2 Report
Comments and Suggestions for Authors
All suggested changes have been performed.
Author Response
All suggested changes have been performed.
-Thanks for the comments.